# Did socioeconomic inequalities in overweight and obesity in South African women of childbearing age improve between 1998 and 2016? A decomposition analysis

**Mweete D. Nglazi** [1,2,3,4] *, **John E. Ataguba** [4,5,6,7]

**1** Implementation Science Centre for Advancing Practice and Training (IMPACT), University of Zambia, Lusaka, Zambia, **2** Department of Clinical Sciences, School of Medicine and Health Sciences, University of Lusaka, Lusaka, Zambia, **3** Department of Community and Family Medicine, School of Public Health, University of Zambia, Zambia, **4** Health Economics Unit, School of Public Health & Family Medicine, University of Cape Town, Cape Town, South Africa, **5** Rady Faculty of Health Sciences, Department of Community Health Sciences, Health Economics Laboratory, Max Rady College of Medicine, University of Manitoba, Winnipeg, Canada, **6** School of Health Systems and Public Health, University of Pretoria, Pretoria, South Africa, **7** Partnership for Economic Policy, Duduville Campus, Kasarani, Nairobi, Kenya

* mweete.nglazi@gmail.com

**Data Availability Statement:** The dataset used for this work are available at the University of Cape Town's DataFirst portal (https://www.datafirst.uct.ac.za/).

## Abstract

Overweight and obesity in adult women contribute to deaths and disability from non-communicable diseases (NCDs) and obesity-related health problems in their offspring. Globally, overweight and obesity prevalence among women of childbearing age (WCBA) has increased, but associated socioeconomic inequality remains unclear. This study, therefore, assesses the changing patterns in the socioeconomic inequality in overweight and obesity among South African non-pregnant WCBA between 1998 and 2016. It uses data from the 1998 and 2016 Demographic and Health Surveys. Socioeconomic inequality in overweight and obesity was assessed using the concentration index ($C$). The index was decomposed to identify contributing factors to obesity and overweight inequalities. Factors contributing to changes in inequalities between 1998 and 2016 were assessed using the Oaxaca-type decomposition approach. Socioeconomic inequalities in overweight and obesity among WCBA in South Africa increased between 1998 ($C$ of 0.02 and 0.06, respectively) and 2016 ($C$ of 0.04 and 0.08, respectively). Socioeconomic status was the biggest contributor to overweight and obesity inequalities for both years. The Oaxaca-type decomposition showed that race and urban residence are major contributors to changes in overweight and obesity inequalities. Policies such as the current tax on sugar-sweetened beverages and subsidising fruits and vegetables, among others, are needed to prioritise WCBA, especially for those from disadvantaged socioeconomic backgrounds, in addressing inequalities in overweight and obesity in South Africa.

**Funding:** The work reported herein was made possible through funding by the South African Medical Research Council (SAMRC) through its Division of Research Capacity Development under the National Health Scholarship Programme from funding received from the Public Health Enhancement Fund/South African National Department of Health. The content hereof is the sole responsibility of the authors and does not necessarily represent the official views of the SAMRC. JEA is supported by the Canada Research Chair. The funders had no role in study design, data collection and analysis, decision to publish, or preparation of the manuscript.

**Competing interests:** The authors have declared that no competing interests exist.

## Introduction

The global prevalence of overweight and obesity among adults increased substantially since 1975, presenting a significant challenge for health and wellbeing [1]. In 2016, over 2 billion adults worldwide were overweight or obese [1], with over 70% living in low- and middle-income countries (LMIC) [2]. Overweight and obesity contribute substantially to deaths and disability from non-communicable diseases (NCDs) [3–7]. Many LMICs facing malnutrition now face the two issues of undernutrition and obesity simultaneously [8]. In Africa [2] and South Africa [6,9–11], in particular, overweight and obesity prevalence has also increased over time, disproportionately affecting women, especially women of childbearing age (WCBA) [7,12]. The burden of overweight and obesity concerns WCBA (i.e., 15 to 49 years old) because it affects their health and can potentially affect the next generation's health [13]. Obesity during a woman's childbearing years is associated with an increased risk of infertility, miscarriage, giving birth to stillbirth children and those with congenital disabilities, shoulder dystocia and other adverse obstetric outcomes [14–19]. WCBA tend to accumulate weight faster during this life stage [20–23]. There are several reasons for the overweight and obesity epidemic occurring during this life stage, with many related to socioeconomic conditions. For example, there are differences between people from wealthier and deprived households in behaviours such as the various types of consumption (smoking, alcohol, processed foods and foods high in fat, salt and sugar) and leisure-time physical activity [24]. Differences in the prevalence of overweight and obesity between socioeconomic groups are partly due to a clustering of health-damaging behaviours by socioeconomic groups [24]. Behaviours such as eating processed foods, foods high in sugar and foods prepared outside the home, for example, are related to individuals' socioeconomic status and associated with an increased risk of obesity and associated NCDs [25,26].

The rapid social and economic development experienced globally in the past four decades has seen a parallel nutrition transition [27,28] with a rise in obesity and NCDs [29], including adult obesity prevalence for both sexes in South Africa [30]. The production and consumption of processed foods, including those with added sugar and salt, increased [31,32], and inequalities in health outcomes, including obesity and its associated social determinants, persist [33–35] in South Africa. Understanding the changes in socioeconomic inequality in overweight and obesity and the associated determinants among WCBA could potentially be one of the ways to address the obesity problem in South Africa through policy.

Although health inequalities literature in South Africa is growing, there is still limited literature concerning the socioeconomic inequalities in overweight and obesity [34,36]. Only one study in South Africa has examined socioeconomic inequality in obesity among adults and found that obesity occurs more frequently among men and women from wealthier households compared to their less affluent counterparts. However, the result for women, even though statistically significant, was marginal (concentration index = 0.09) compared to that for men (0.12), mainly due to obesity's high prevalence among women [34]. That study was considered only a single year and focused on obesity among men and women. It did not assess inequalities in overweight and the factors that explain changes in obesity and overweight inequality in South Africa. This paper aims to assess, for the first time, the changing patterns in the socioeconomic inequality in overweight and obesity among South African non-pregnant WCBA between 1998 and 2016. It also assesses the factors that explain the changes in socioeconomic inequalities in overweight and obesity.

## Methods

### Ethics statement

This paper uses publicly available SADHS data that have received ethics approval. The SADHS are nationally representative cross-sectional household surveys conducted by the National Department of Health, Statistics South Africa, South African Medical Research Council and ICF International. Although secondary data are used in this paper, ethics approval was also received from the Human Research Ethics Committee at the University of Cape Town (HREC Reference 409/2019).

### Data sources

Data were obtained from the nationally representative South Africa Demographic and Health Surveys (SADHS) for 1998 and 2016, which are publicly available at https://dhsprogram.com/data/available-datasets.cfm. This paper did not use the 2003 SADHS data because they are not available in any public data repository [37] and are not available to researchers. The 1998 (adult survey) and 2016 (women's survey) SADHS, which formed the major dataset used in this paper, had a total sample size of 8,074 and 8,514 women, respectively [38,39], with the multistage sampling procedures detailed elsewhere [38,39]. The SADHS collect information from a household questionnaire, biomarker questionnaire, woman's questionnaire and a man's questionnaire with data stored in several dataset files: household recode, individual recode, birth recode, kids recode, men's recode, and couples recode. In this paper, the dataset containing data on women was complemented with household-specific data, like wealth, from the household file. S1 Fig presents a short flowchart describing the data-cleaning process used to arrive at the final WCBA sample for 1998 and 2016.

Table 1 describes the key variables used in this paper. Variable selection was based on the Dahlgren and Whitehead model [40] concerning important social determinants of obesity and overweight and availability across the SADHS datasets. Socioeconomic status was proxied using the wealth index created within the SADHS data using a method described by Rutstein and Johnson [41]. Although the wealth index in the SADHS is essentially a measure of household economic or wealth status, it is important to note that socioeconomic status is broader than household economic or wealth status. The wealth index was used to generate national wealth or socioeconomic quintiles.

### Statistical analysis

**Descriptive statistics.** Descriptive statistics (means for continuous variables and proportions for categorical variables) were used to summarise the data after accounting for the sampling design and applying the appropriate sample weights to obtain national figures.

### Analytical methods for estimating health inequality

**Concentration index.** The concentration index (*C*) [44] was used to assess socioeconomic inequality in overweight and obesity. The standard *C* was computed via the "convenient regression" Equation [45]:

$$2\sigma_r^2\left(\frac{h_i}{\mu_h}\right) = \alpha + Cr_i + \epsilon_i \tag{1}$$

where *C* is the concentration index for overweight or obesity ($h_i$), $\mu_h$ is the mean of *h* or the proportion of overweight and obese WCBA, $r_i$ is the fractional rank of a woman in the living

**Table 1. A description of key variables used in the analysis.**

| Variable | Definition |
| --- | --- |
| *Variables of interest* | |
| Overweight | A body mass index (BMI) $\geq$ 25 kg/m$^2$ [42] |
| Obesity | A BMI $\geq$ 30 kg/m$^2$ [42] |
| *Determinants* | |
| Age | A woman's age in years |
| Black African[1] | A dummy variable for women who self-identified as black African race |
| Coloured | A dummy variable for women who self-identified as coloured |
| Indian/Asian | A dummy variable for women who self-identified as Indian/Asian race |
| White | A dummy variable for women who self-identified as white |
| No schooling/ | A dummy variable for a woman with no formal education |
| primary education | A dummy variable for a woman attaining only primary education |
| Secondary | A dummy variable for a woman with secondary education |
| Tertiary education | A dummy variable for a woman with tertiary education |
| Employed | A dummy variable for a woman holding formal or informal employment |
| Unemployed | A dummy variable for an unemployed woman |
| Married/living together | A dummy variable for a woman who is currently married or living together with a partner |
| Single/never married | A dummy variable for a single or never-married woman |
| Widowed or divorced | A dummy variable for a woman who is widowed, separated or divorced |
| Rural | A woman residing in a rural location |
| Urban | A woman residing in an urban location |
| Smoking | A woman who reported currently smoking |
| Quintiles of socioeconomic status (Quintiles 1–5)[2] | Quintile 1 = 1 if a woman is in the poorest socioeconomic group; 0 otherwise<br>Quintile 2 = 1 if a woman is in the second poorest socioeconomic group; 0 otherwise<br>Quintile 3 = 1 if a woman is in the middle socioeconomic group; 0 otherwise<br>Quintile 4 = 1 if a woman is in the second richest socioeconomic group; 0 otherwise<br>Quintile 5 = 1 if a woman is in the richest socioeconomic group; 0 otherwise |

Notes:

[1]The South African population is predominantly black, and racial disparities have been reported for obesity and overweight [43];

[2]Quintiles of socioeconomic status are based on the household wealth index for the Demographic and Health Survey data.

standards distribution based on the wealth index, $\sigma_r^2$ is the variance of the fractional rank, $\alpha$ is the intercept, and $\epsilon_i$ is the error term [45,46].

The theoretical value of the $C$ ranges from -1 to +1. A positive concentration index ($C > 0$) denotes a pro-rich distribution where obese or overweight women are more likely to be wealthier than their counterparts. In contrast, a negative index ($C < 0$) denotes a pro-poor distribution and signifies the opposite. It has been suggested to normalise the concentration index when the variable of interest is binary, like "obesity" and "overweight," as used in this paper, to ensure that the theoretical value of $C$ remains in the [-1,1] range [47]. However, the approaches suggested in the literature [47,48] for normalising and adjusting the concentration

index may produce results counterintuitive for policy to reduce the burden of overweight and obesity among the most affected populations [49]. Therefore, this paper did not normalise the concentration index but presents the results for the standard concentration index [49]. Further, the concentration index may be standardised for age/sex variations or key explanatory variables [35,46]. This adjustment was unnecessary in this paper as the decomposition analysis that follows using these variables is a detailed approach to assessing and explaining key factors contributing to the concentration index.

**Decomposing the concentration index.** The $C$ obtained in Eq 1 was decomposed to explain the drivers (i.e., the contributions of the determinants listed in Table 1) of socioeconomic inequalities in obesity and overweight among non-pregnant WCBA. Therefore, let the relationship between the indicator of overweight or obesity ($h$) and the $k$ explanatory variables ($x_k$) be given in a regression model [50] as:

$$h_i = \alpha + \sum_k \beta_k x_{ki} + \varepsilon_i \tag{2}$$

where $\alpha$ and $\beta$ are parameters, with $\beta$ measuring the relationship between each explanatory factor ($k$), while $\varepsilon$ denotes the error term.

The concentration index, $C$, obtained from Eq 1 can be re-written, taking into account the relationship in Eq 2 as [50]:

$$C = \sum_k (\beta_k \bar{x}_k / \mu_h) C_k + GC_\varepsilon / \mu_h \tag{3}$$

where $h$ and $\mu_h$ remain as previously described, $\bar{x}_k$ is the mean of each explanatory factor ($k$), $\beta_k$ is the parameter estimated from Eq 2 for each explanatory factor, $C_k$ denotes the concentration index for the $k$-th contributing factor, while $GC_\varepsilon$ is the generalised concentration index for the error term ($\varepsilon$) in Eq 2.

The contribution of each explanatory factor to the concentration index for inequality in overweight or obesity (i.e.,$(\beta_k \bar{x}_k / \mu_h) C_k$) is the product of the concentration index of each factor, $C_k$, and the elasticity of $h$ with respect to each explanatory factor ($\beta_k \bar{x}_k / \mu_h$). The last component in Eq 3 ($GC_\varepsilon / \mu_h$) is socioeconomic inequality in overweight or obesity that variations in the explanatory factors across socioeconomic groups cannot systematically explain. Generally, when the concentration index for overweight or obesity is positive, a positive value for any explanatory factor means that the factor contributes to the concentration of inequality in overweight or obesity among WCBA from wealthier households.

**Decomposing changes in the concentration index.** Changes in socioeconomic inequalities between two time periods ($\Delta C = C_t - C_{t-1}$) could be either pro-poor or pro-rich, with a pro-rich change or shift meaning that inequality "favours" those from wealthier backgrounds [51]. In this paper, the factors explaining and contributing to changes in socioeconomic inequalities in overweight and obesity between 1998 and 2016 were assessed using the Oaxaca-type decomposition approach [50] via the formula:

$$\Delta C = C_t - C_{t-1} = \sum_k \eta_{kt}(C_{kt} - C_{kt-1}) + \sum_k C_{kt-1}(\eta_{kt} - \eta_{kt-1}) + \Delta\left(\frac{GC_{\epsilon t}}{v_t}\right) \tag{4}$$

The subscripts $t$ and $t - 1$ represent the years 2016 and 1998, respectively. $\Delta$ denotes the first differences and $\eta_k$ is the elasticity of $h$ with respect to $\bar{x}_k$ (i.e.,$\eta_k = \beta_k \bar{x}_{ki}$).

Three decomposition approaches may be used, namely the Oaxaca-type method, the total differential approach [50] and a recent method proposed by Ataguba and colleagues [51] that focuses on inequalities between- and within-socioeconomic groups. The Oaxaca-type decomposition approach used in this paper decomposes the change in socioeconomic inequality in

overweight and obesity over time (i.e., between two time periods denoted by subscripts $t$ and $t − 1$) into changes in the concentration index ($C_{kt}$ and $C_{kt−1}$) and elasticities ($\eta_{kt}$ and $\eta_{kt−1}$) of the factors explaining and contributing to changes in socioeconomic inequalities in overweight and obesity. For each contributing factor or variable, changes in the elasticities between two time periods are weighted by the concentration index and the changes in the concentration index by the elasticities, as shown in Eq 4. One limitation of the Oaxaca-type method was the difficulty disentangling changes within elasticities [50], which the total differential approach (TDA) presents as an alternative. Because this paper was not intended to unpack changes within elasticities, which is of limited policy relevance, it uses the Oaxaca-type decomposition approach widely used in applied research. Further, the total differential approach is only applicable for small changes in inequality [50], which makes its application rather restrictive compared to the Oaxaca-type approach that is encompassing. Also, the Oaxaca-type decomposition has the added advantage of being easily understood. The recently proposed decomposition framework by Ataguba and colleagues [51] decomposes changes in socioeconomic inequalities into changes between- and within-socioeconomic group health inequalities. It does not show variables or factors that explain changes in socioeconomic inequalities, a major focus of this paper.

Data analysis was done using Stata v18 [52]. The standard errors for Eqs 3 and 4 components were obtained using the bootstrap methods with 1,000 replications [53,54].

## Results

### Summary statistics

The sample size for the final analysis was 4,939 WCBA in the 1998 SADHS and 3,144 in the 2016 SADHS (Table 2). The prevalence of overweight in WCBA increased from 50.3% to 62.0% between 1998 and 2016, while the prevalence of obesity increased from 24.3% to 35.2%. In both periods, the average age of WCBA was about 30 years. The 1998 and 2016 SADHS data had more self-identified Black African WCBA than other race groups. Compared with the 1998 SADHS, the 2016 SADHS had more WCBA attaining secondary and tertiary education but fewer WCBA attaining primary education or without formal schooling. Besides, more WCBA resided in urban areas than rural areas during both survey years. The prevalence of smoking among WCBA decreased from 11.5% to 4.4% between 1998 and 2016.

### Concentration indices for 1998 and 2016

Socioeconomic inequality in overweight and obesity among WCBA was pro-rich, with the positive concentration index increasing from 0.02 (in 1998) to 0.04 (in 2016) for overweight and from 0.06 (in 1998) to 0.08 (in 2016) for obesity. These pro-rich distributions are statistically significant except for the 1998 concentration index of overweight (0.02). Pro-rich inequalities indicate that WCBA from wealthier households are likelier to be obese and overweight than their less wealthy counterparts. Moreover, because the positive concentration indices for 2016 were greater than those for 1998, overweight and obesity inequalities became more pro-rich, increasing their concentration among WCBA from wealthier households between 1998 and 2016.

### Decomposition results

The contributions of different factors to explaining socioeconomic inequalities in overweight and obesity in 1998 and 2016 are shown in Fig 1. A positive sign on a contributing factor denotes that, *ceteris paribus*, the positive socioeconomic inequality in overweight or obesity

**Table 2. Summary statistics of the sample of women of childbearing age (15 and 49 years old) for 1998 and 2016.**

| | Survey year | |
|---|---|---|
| | **1998** | **2016** |
| Sample | 4,939 | 3,144 |
| Overweight | 50.3 (48.6–52.1) | 62.0 (59.6–64.3) |
| Obese | 24.3 (22.8–25.9) | 35.2 (32.8–37.6) |
| Age, mean (Standard deviation) | 29.2 (9.7) | 30.2 (9.9) |
| *Race* | | |
| Black African | 77.6 (75.0–79.9) | 90.6 (88.4–92.4) |
| Coloured | 10.2 (8.8–11.9) | 5.8 (4.7–7.2) |
| Asian/Indian | 4.2 (3.1–5.5) | 1.5 (0.7–3.1) |
| White | 8.0 (6.5–9.9) | 2.1 (1.3–3.4) |
| *Education* | | |
| No schooling | 6.5 (5.5–7.6) | 2.2 (1.6–3.0) |
| Primary | 33.9 (32.1–35.7) | 10.0 (8.7–11.6) |
| Secondary | 52.4 (50.4–54.4) | 77.6 (75.4–79.6) |
| Tertiary | 7.3 (6.3–8.4) | 10.2 (8.4–12.2) |
| *Employment status* | | |
| Employed | 37.1 (35.0–39.3) | 33.0 (30.6–35.5) |
| Unemployed | 62.9 (60.7–65.0) | 67.0 (64.5–69.4) |
| *Marital status* | | |
| Married/living together | 42.3 (40.4–44.2) | 33.8 (31.1–36.5) |
| Single/never married | 53.2 (51.3–55.1) | 63.3 (60.7–65.9) |
| Widowed or divorced | 4.5 (3.9–5.2) | 2.9 (2.3–3.8) |
| *Area of residence* | | |
| Rural | 37.0 (35.3–38.7) | 36.7 (33.8–39.7) |
| Urban | 63.0 (61.3–64.7) | 63.3 (60.3–66.2) |
| *Lifestyle* | | |
| Smoking | 11.5 (10.2–12.9) | 4.4 (3.5–5.4) |
| *Socioeconomic status quintile* | | |
| Q1 (poorest) | 15.2 (13.3–17.2) | 21.8 (18.5–25.4) |
| Q2 | 18.3 (16.5–20.4) | 20.2 (17.9–22.8) |
| Q3 | 20.2 (18.3–22.2) | 22.8 (20.1–25.7) |
| Q4 | 23.2 (21.0–25.6) | 19.6 (17.0–22.6) |
| Q5 (richest) | 23.1 (20.8–25.6) | 15.6 (12.8–19.0) |

Notes

a) Summary statistics reported in percentages except for age reported as mean

b) Estimates are weighted to the population using the SADHS full sampling design

c) Standard deviation or 95% logit-transformed confidence interval are displayed in parenthesis.

would be reduced if that factor was absent or had no contribution (the opposite for a negative sign on a contributing factor).

In 1998, as shown in Fig 1, the pro-rich concentration index of overweight was due in part to positive contributions of socioeconomic status (approx. +20%), age (approx. +15%) and urban residence (approximately +15%); and negative contribution of race (approx. -30%). In 2016, most of the socioeconomic inequality in overweight was explained by socioeconomic status (approx. +55%), age (approx. +15%) and race (approx. -12%). In both 1998 and 2016, the concentration indices of obesity are due in part to the positive contributions of socioeconomic

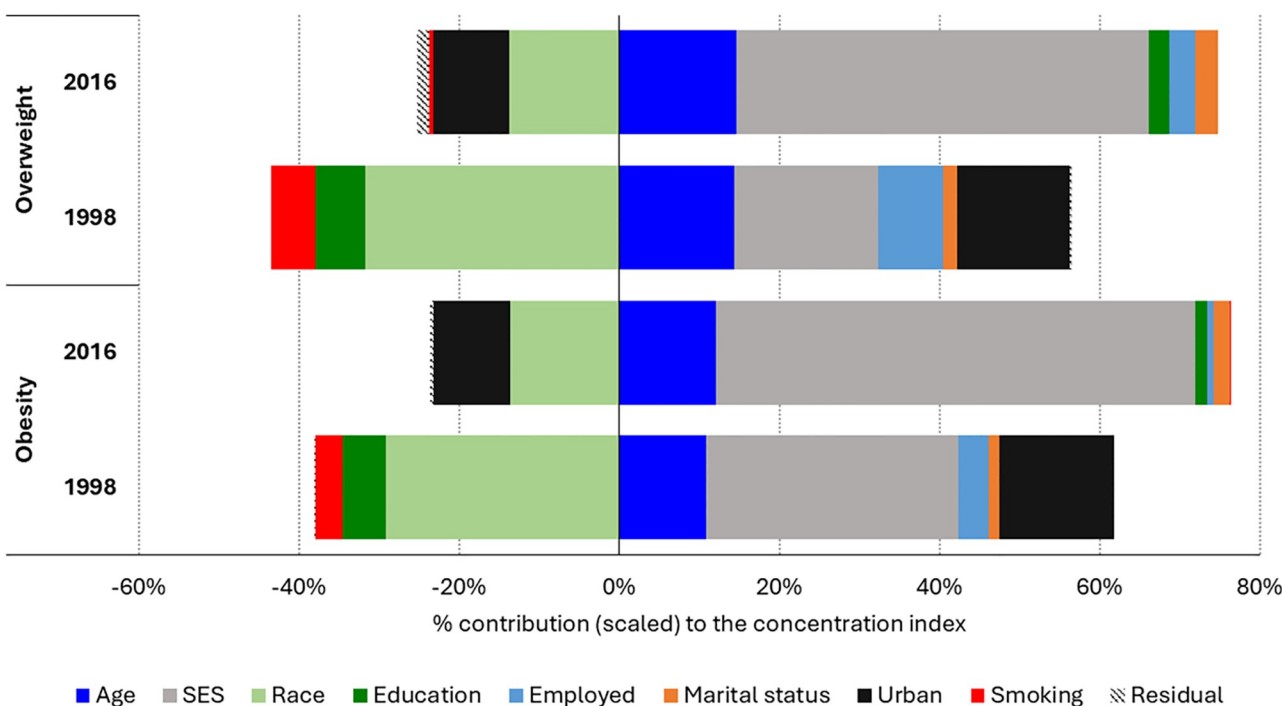

**Fig 1. Relative contribution of factors explaining socioeconomic inequality in overweight and obesity among women of childbearing age (15 and 49 years) for 1998 and 2016, South Africa.**

status (approx. +32% in 1998 and +60% in 2016) and age (+10% in 1998 and 12% in 2016); and the negative contributions of education (approx. -29% in 1998 and -12% in 2016). The contribution of urban residence to socioeconomic inequality in obesity among WCBA was positive (approx. 14%) in 1998 but negative in 2016 (approx. -10%). Detailed analyses of the contributions of the different factors in explaining socioeconomic inequality in overweight and obesity are presented in the S1 and S2 Tables.

The decomposition of changes in the socioeconomic inequality in overweight and obesity among women between 1998 and 2016 (Fig 2) shows the percentage contribution of each contributing factor to the observed changes in overweight and obesity inequalities. A positive percentage means that the factor contributes to an increase in socioeconomic inequality in overweight or obesity among WCBA from wealthier households (i.e., inequality is becoming more pro-rich or more concentrated among WCBA from wealthier households) between 1998 and 2016. A negative percentage indicates that the factor contributes to reducing socioeconomic inequalities among WCBA from wealthier households. The results in Fig 2 show race as the most significant contributor to changes in socioeconomic inequalities in overweight (approx. +29%) and obesity (+34%) between 1998 and 2016, considering the entire population of WCBA. Although negative, urban residence was also a prominent contributor to changes in socioeconomic inequalities in overweight (approx. -25%) and obesity (approx. -31%) for the overall population of WCBA. Socioeconomic status and educational attainment contributed to the pro-rich inequalities in overweight between 1998 and 2016 in the overall population of WCBA. The results of the urban/rural sub-population analyses in Fig 2 show a similar pattern found in the overall population of WCBA for both overweight and obesity. Prominent factors explaining the positive changes in the concentration indices for obesity and overweight among WCBA in rural and urban areas include socioeconomic status, race and education attainment.

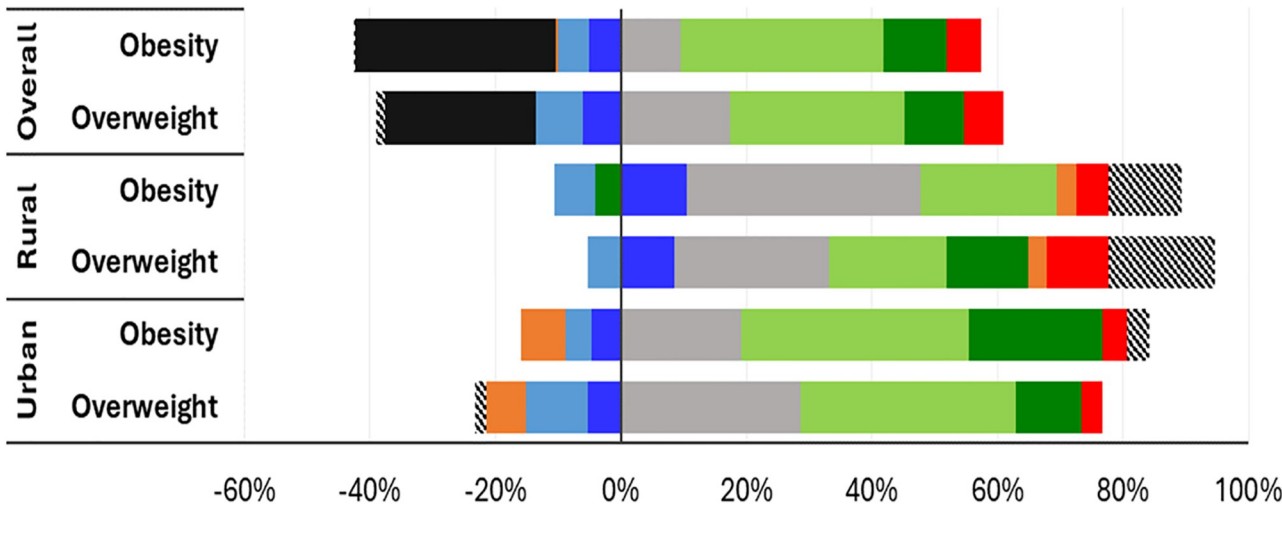

**Fig 2. Relative contribution of factors explaining changes in overweight and obesity inequalities among women of childbearing age (15 and 49 years) for 1998 and 2016, South Africa.**

Minor differences exist between the urban and rural results. Age and marital status were marginally positive contributors to changes in the socioeconomic inequalities in overweight and obesity in rural areas but positive in urban areas. Also, the residual components were more pronounced for the rural analysis than for the overall and urban sub-analyses. S3 Table contains the detailed decomposition of changes in the concentration indices for overweight and obesity.

## Discussion

This study investigated the socioeconomic inequality in overweight and obesity in South Africa among WCBA between 1998 and 2016. WCBA from wealthier households bear a higher burden of obesity and overweight than their less wealthier counterparts, which increased between 1998 and 2016. The main contributors to socioeconomic inequality in overweight and obesity were socioeconomic status, age and urban residence. While the contributions of race groups and urban residence were more prominent for the changes in socioeconomic inequality in overweight and obesity among WCBA, SES and race were the most prominent for the urban/rural sub-analysis.

Prior research indicates mixed findings regarding socioeconomic inequalities in overweight and obesity. While some studies found overweight and obesity occurring more among people from wealthier backgrounds [55–60], others reported the opposite [55,61–64]. There are gender differences in socioeconomic inequalities in overweight and obesity [34,63]. Evidence from Spain [61], Canada [55], Iran [62] and Brazil [60] shows significant socioeconomic inequality, with obesity and/or overweight occurring more among poorer populations (women, men or the general adult population) than their wealthier counterparts. There was also evidence that overweight or obesity could occur more among wealthier population

groups, as found in Canada and South Africa, where obesity occurred more among men from wealthier backgrounds [34,55], and in Iran, where overweight and obesity occurred more among wealthier adults [58,59]. Also, as reported in South Africa, obesity could be more evenly spread among women, irrespective of their socioeconomic class, compared to men [34]. Significant factors that explain socioeconomic inequalities in overweight and obesity, especially in the general population, include wealth, alcohol consumption and smoking habits, occupational status, educational attainment, residency status, sedentary lifestyle, physical inactivity, and marital status [34,55,56,58–60,62]. A study in Bangladesh specifically examined socioeconomic inequality in overweight and obesity among non-pregnant WCBA, with overweight and obesity occurring more among women from wealthier backgrounds [56].

Fewer studies, predominantly from high-income countries, estimate and decompose changes in socioeconomic inequality in overweight and obesity over time. For example, changes in socioeconomic inequality among Swedish adults between 1980 and 1997 show that obesity was increasingly concentrated among males and females from wealthier backgrounds than their counterparts from socioeconomically deprived backgrounds [57]. The study among Korean (in 1998 and 2015) among adults showed mixed results, with obesity becoming more concentrated among women from poorer backgrounds but among men from wealthier backgrounds [63]. Significant factors explaining changes in socioeconomic inequalities in obesity include key sociodemographic factors like age, educational attainment and income [57,63].

This study contributes novel insights by utilising nationally representative datasets to examine the changing patterns in the socioeconomic inequality in overweight and obesity among South African non-pregnant WCBA between 1998 and 2016. Overweight and obesity prevalence among WCBA increased in South Africa, but associated socioeconomic inequality remained unclear. To the best of our knowledge, this study was the first to report on the changing patterns in the socioeconomic inequality in overweight and obesity among South African non-pregnant WCBA (and indeed in Africa) and factors explaining these changes in socioeconomic inequality between 1998 and 2016. This study uses the concentration index and decomposition techniques to estimate the socioeconomic inequality in overweight and obesity, identify the factors that explain the inequality in 1998 and 2016, and determine the factors explaining the change in socioeconomic inequality between 1998 and 2016. Overweight and obesity occur more among WCBA from wealthier backgrounds in South Africa, consistent with a previous South Africa study among men and women [34]. The previous study in South Africa found that although obesity prevalence was higher among women than men, the concentration was relatively even across socioeconomic groups for women compared to men [34]. Our paper, focusing on non-pregnant WCBA, reported concentration indices similar to those reported previously for women in general, meaning that there are few differences between the general adult female population and the non-pregnant WCBA in terms of socioeconomic inequalities in obesity. It is important to note that the previous paper focused solely on obesity but did not assess socioeconomic inequalities in overweight, with a higher prevalence than obesity. Also, the previous study did not disaggregate inequalities by rural/urban location.

In keeping with previous studies [34,56], we also find that socioeconomic status was a major contributor to socioeconomic inequality in overweight and obesity among WCBA in 1998 and 2016. Age also plays an important role in explaining the socioeconomic inequality in overweight and obesity among WCBA in 1998 and 2016. In the context of age, the critical reproductive period often leads to nonlinearities in accumulating inequality [65]. Also, we find that urbanisation, and likely its accompanying stress [66], was one of the main contributors to obesity inequality. For example, Bartley [67] describes how feelings of inequality, domination or subordination can stimulate stress responses in the body, adversely affecting health and health outcomes. Social differences in health could result from an unequal distribution of

psychological risk factors. Factors such as the stress of urbanisation may increase the incidence of chronic diseases of lifestyle (including female obesity). For instance, Steyn and colleagues [66] investigated urban exposure among 986 men and women aged 15 to 64 years who self-identified as Black Africans, living in the Cape Peninsula, South Africa, concerning unhealthy lifestyles and risk factors for chronic diseases. They found that those who spent more extensive portions of their lives in an urban setting tended to have unhealthier lifestyles and a higher risk for chronic diseases of lifestyle (including female obesity) than their less urbanised counter-parts [66]. In urban areas, the results of this study show race as a major contributor to the changes in socioeconomic inequality in obesity and overweight compared to socioeconomic status as a major contributor in rural areas. Besides, education was one of the main contributors to socioeconomic inequality in obesity among WCBA in 1998 and 2016 as reported elsewhere [56].

By examining the changes in socioeconomic inequality in overweight and obesity, we find a pro-rich shift [51] between 1998 and 2016 (i.e., overweight and obesity increasingly being concentrated among women from wealthier backgrounds) for WCBA in South Africa. This was consistent for urban and rural locations. Oaxaca decomposition results showed that the significant contributors to changes in socioeconomic inequality in overweight and obesity between 1998 and 2016 were race (self-identified as black), urban residence and lifestyle (smoking). Employment status was also significant for changes in overweight inequality but not for obesity. While it is difficult to explain the role of race in this study, previous research found the self-identified Black African population group perceiving weight gain as a sign of beauty, wealth and being healthy [68]. Women without formal educational qualifications tend to be more physically inactive [11]. Also, higher-educated women are employed with better-paying jobs and choose more expensive but not always healthier food products since consuming such items might be a sign of prestige [69]. Because smoking is more prevalent among women from wealthier households (the concentration indices for smoking computed in this study were positive), reducing smoking incidence, especially among wealthier populations, will reduce the pro-rich changes in obesity and overweight inequality among WCBA in South Africa.

Despite its contributions, the study has a few limitations. One limitation was the assumption of linearity in estimating the determinants of obesity and overweight, which were binary indicators. However, as noted elsewhere [70], this assumption does not change the qualitative results from decomposing socioeconomic inequality in overweight or obesity. Also, due to data availability constraints, this paper used the wealth index in the SADHS, a measure of household economic status, to proxy broader socioeconomic status. The study has some strengths. Comparable nationally representative data are used to provide a picture of the entire country after accounting for sampling weights. Also, heights and weights are used to compute objectively measured BMI values. Because pregnancy can inflate BMI, only non-pregnant women are used in this study.

## Conclusion

Women from all socioeconomic backgrounds, especially those from poorer backgrounds, should be prioritised to reduce disparities in overweight and obesity among WCBA in South Africa, recognising the social determinants of health inequality reported in this paper. Cost-effective interventions to tackle overweight and obesity include the current tax on sugar-sweetened beverages [71], reducing taxes or subsidising fruit and vegetables, promoting food labelling and regulating food formulation [2] addressing the availability and affordability of fruits and vegetables [72,73], even though they are zero-rated from value-added tax.

## Supporting information

**S1 Fig. Data cleaning flowchart.**
(TIF)

**S1 Table. Decomposition of the concentration index for overweight among women of childbearing age 15–49 years, South Africa, 1998 and 2016.**
(DOCX)

**S2 Table. Decomposition of the concentration index for obesity among women of childbearing age (15–49 years), South Africa, 1998 and 2016.**
(DOCX)

**S3 Table. Decomposition of change in concentration index for overweight and obesity among women of childbearing age (15–49 years), South Africa, 1998–2016.**
(DOCX)

## Author Contributions

**Conceptualization:** John E. Ataguba.

**Data curation:** Mweete D. Nglazi.

**Formal analysis:** Mweete D. Nglazi, John E. Ataguba.

**Methodology:** Mweete D. Nglazi, John E. Ataguba.

**Visualization:** Mweete D. Nglazi, John E. Ataguba.

**Writing – original draft:** Mweete D. Nglazi.

**Writing – review & editing:** Mweete D. Nglazi, John E. Ataguba.

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
