## [Decision Letter · Decision Letter 0]

12 Jul 2024

PGPH-D-24-00977

Did socioeconomic inequalities in overweight and obesity in South African women of childbearing age improve between 1998 and 2016? A decomposition analysis

Dear Dr. Nglazi,

Thank you for submitting your manuscript to PLOS Global Public Health. After careful consideration, we feel that it has merit but does not fully meet PLOS Global Public Health’s publication criteria as it currently stands. Therefore, we invite you to submit a revised version of the manuscript that addresses the points raised during the review process.

Please submit your revised manuscript by . If you will need more time than this to complete your revisions, please reply to this message or contact the journal office at globalpubhealth@plos.org. Please include the following items when submitting your revised manuscript:

We look forward to receiving your revised manuscript.

Kind regards,

Sanmei Chen

Academic Editor

Journal Requirements:

Additional Editor Comments (if provided):

1. The manuscript would benefit from a more detailed explanation of how the concentration index was adjusted for relevant variables. Additionally, it is essential to provide a clearer rationale for choosing the Oaxaca Decomposition Approach in the context of this study. Please elaborate on these points to enhance the methodological transparency and rigor.

2. To improve clarity and replicability, we recommend including a flowchart detailing the process of assembling the final sample. Furthermore, please provide explicit information regarding the inclusion and exclusion criteria used for the analysis. These additions will enhance the comprehensibility of the methodology section.

3. It is crucial to note that the household wealth index in the DHS survey serves as an indicator of household economic status rather than directly reflecting socioeconomic status. Clarifying this distinction will ensure the accurate interpretation of results related to socioeconomic disparities.

4. Consideration may be given to including marriage and employment as determinants in the analysis. If not included, please provide a rationale for their exclusion to strengthen the conceptual framework of the study.

5. The manuscript would benefit from improvements in English writing quality.

6. Please ensure that the manuscript preparation strictly follows the submission guidelines outlined by the journal, available at https://journals.plos.org/globalpublichealth/s/submission-guidelines

Reviewers' comments:

Reviewer's Responses to Questions

**Comments to the Author**

1. Does this manuscript meet PLOS Global Public Health’s publication criteria? Is the manuscript technically sound, and do the data support the conclusions? The manuscript must describe methodologically and ethically rigorous research with conclusions that are appropriately drawn based on the data presented.

Reviewer #1: No

Reviewer #2: Yes

Reviewer #3: Yes

2. Has the statistical analysis been performed appropriately and rigorously?

Reviewer #1: No

Reviewer #2: Yes

Reviewer #3: Yes

3. Have the authors made all data underlying the findings in their manuscript fully available (please refer to the Data Availability Statement at the start of the manuscript PDF file)?

Reviewer #1: No

Reviewer #2: Yes

Reviewer #3: Yes

4. Is the manuscript presented in an intelligible fashion and written in standard English?

Reviewer #1: No

Reviewer #2: Yes

Reviewer #3: Yes

5. Review Comments to the Author

Reviewer #1: This submission is not new. The authors inadvertently selected the option for a 'new submission' instead of submitting a revised version. I am managing the original manuscript.

xxxxxxxxxxxxxxxx

Reviewer #2: This is a well written article. I just have few comments that will make the article clearer to the readers:

1. Concentration Index Adjustment: The concentration index (CI) is a widely utilized measure in socioeconomic inequality analysis. However, it is crucial to address whether the CI has been adjusted to account for any biases or variations that could affect the results. To ensure clarity, the authors should include a detailed statement in the methodology section, specifying if the CI was adjusted. If adjustments were made, the authors should explain the rationale and the methods used to adjust the CI. Conversely, if no adjustment was made, the authors should provide a clear explanation for this decision

2. The Oaxaca Decomposition Approach: The article currently presents the use of the Oaxaca decomposition approach, but the rationale behind its selection remains unclear. To enhance the readers' comprehension, the authors should elaborate on why this specific method was chosen. They should provide a brief explanation of the Oaxaca decomposition approach, highlighting its relevance and advantages in the context of their study. This expanded explanation in the methodology section will offer a thorough understanding of the approach and its application in analyzing the data.

Reviewer #3: The manuscript has strengths but some of the issues needs to be addressed

The authors have well presented the Oaxaca Decomposition approach used for socio-economic inequalities in overweight and obesity among women of child bearing age.

Some typos need to corrected in preliminary section (Competing interests) and Data availability (review the journal criteria- the data needs to be provided in a supplementary file)

Rationale set well for the study

Socioeconomic status was biggest contributor for both years

Disaggregated results on age groups could be presented as it has covered a large range of age group.

Race was major contributor to change in both parameters

Recommendation on abstract needs to be revised based on results, Please revise as you have presented in the conclusion of main text (ie. not for all socioeconomic backgrounds; recommend on specific actions that could be taken for addressing the inequality gap)

Methods: line 3 to 4 could be rewritten as; Data was obtained from the nationally representative South Africa Demographic and Health Surveys (SADHS) of the years 1998 and 2016 which are publicly available at

In section Decomposing changes in the concentration index inequalities over 1998 and 2006 has been quoted I hope this is a typo as authors are comparing with the years 1998 and 2016

There are observed differences in sample sizes quoted by authors in different sections (data sources and summary statistics) which needs to reconfirm and correction. If needed analysis should be re-conducted based on the sample size (I hope it is not the issue).

Analysis of WCBA residing in urban/rural areas vs education status could provide more precise relation on observed in-equalities. Do consider if possible.

Re-Formatting of tables according to journal criteria needs to be done

Further discussion required for explaining possible reasons behind the differences observed with the current study and the findings of reference 33 who reported that obesity could be evenly spread between poorer and richer women which is contrasting in current study. Authors could further discuss on observed changes in these years which can have contributed to this difference.

6. PLOS authors have the option to publish the peer review history of their article (what does this mean?). If published, this will include your full peer review and any attached files.

**Do you want your identity to be public for this peer review?** For information about this choice, including consent withdrawal, please see our Privacy Policy.

Reviewer #1: No

Reviewer #2: No

Reviewer #3: No

---

## [Decision Letter · Decision Letter 1]

10 Oct 2024

PGPH-D-24-00977R1

Did socioeconomic inequalities in overweight and obesity in South African women of childbearing age improve between 1998 and 2016? A decomposition analysis

Dear Dr. Nglazi,

Thank you for submitting your manuscript to PLOS Global Public Health. After careful consideration, we feel that it has merit but does not fully meet PLOS Global Public Health’s publication criteria as it currently stands. Therefore, we invite you to submit a revised version of the manuscript that addresses the points raised during the review process.

We look forward to receiving your revised manuscript.

Kind regards,

Sanmei Chen

Academic Editor

Journal Requirements:

Additional Editor Comments (if provided):

In the S1 Figure: Data cleaning flowchart, "Excluding pregnant women, women aged more than 49 years and observations without a complete set of covariates used for data analysis"

Please include the number of participants excluded for each of the following reasons in the followchart: pregnant women, women over the age of 49, and lacking a complete set of covariates used for data analysis.

Reviewers' comments:

Reviewer's Responses to Questions

**Comments to the Author**

1. If the authors have adequately addressed your comments raised in a previous round of review and you feel that this manuscript is now acceptable for publication, you may indicate that here to bypass the “Comments to the Author” section, enter your conflict of interest statement in the “Confidential to Editor” section, and submit your "Accept" recommendation.

Reviewer #3: All comments have been addressed

2. Does this manuscript meet PLOS Global Public Health’s publication criteria? Is the manuscript technically sound, and do the data support the conclusions? The manuscript must describe methodologically and ethically rigorous research with conclusions that are appropriately drawn based on the data presented.

Reviewer #3: Yes

3. Has the statistical analysis been performed appropriately and rigorously?

Reviewer #3: Yes

4. Have the authors made all data underlying the findings in their manuscript fully available (please refer to the Data Availability Statement at the start of the manuscript PDF file)?

Reviewer #3: Yes

5. Is the manuscript presented in an intelligible fashion and written in standard English?

Reviewer #3: (No Response)

6. Review Comments to the Author

Reviewer #3: No further comments

7. PLOS authors have the option to publish the peer review history of their article (what does this mean?). If published, this will include your full peer review and any attached files.

**Do you want your identity to be public for this peer review?** For information about this choice, including consent withdrawal, please see our Privacy Policy.

Reviewer #3: **Yes: **Rabindra Bhandari

---

## [Editor Report · Decision Letter 2]

29 Oct 2024

Did socioeconomic inequalities in overweight and obesity in South African women of childbearing age improve between 1998 and 2016? A decomposition analysis

PGPH-D-24-00977R2

Dear Dr Nglazi,

We are pleased to inform you that your manuscript 'Did socioeconomic inequalities in overweight and obesity in South African women of childbearing age improve between 1998 and 2016? A decomposition analysis' has been provisionally accepted for publication in PLOS Global Public Health.

Best regards,

Sanmei Chen

Academic Editor